# The Fusion of Lipid and DNA Nanotechnology

**DOI:** 10.3390/genes10121001

**Published:** 2019-12-03

**Authors:** Es Darley, Jasleen Kaur Daljit Singh, Natalie A. Surace, Shelley F. J. Wickham, Matthew A. B. Baker

**Affiliations:** 1School of Biotechnology and Biomolecular Science, UNSW Sydney, Kensington 2052, Australia; s.darley@student.unsw.edu.au; 2School of Chemistry, University of Sydney, Camperdown 2006, Australia; jdal9865@uni.sydney.edu.au (J.K.D.S.); nsur9983@uni.sydney.edu.au (N.A.S.); 3School of Chemical and Biomolecular Engineering, University of Sydney, Camperdown 2006, Australia; 4Sydney Nanoscience Institute, University of Sydney, Camperdown 2006, Australia; 5School of Physics, University of Sydney, Camperdown 2006, Australia; 6CSIRO Synthetic Biology Future Science Platform, GPO Box 2583, Brisbane, QLD 4001, Australia

**Keywords:** lipid nanotechnology, DNA nanotechnology, DNA origami

## Abstract

Lipid membranes form the boundary of many biological compartments, including organelles and cells. Consisting of two leaflets of amphipathic molecules, the bilayer membrane forms an impermeable barrier to ions and small molecules. Controlled transport of molecules across lipid membranes is a fundamental biological process that is facilitated by a diverse range of membrane proteins, including ion-channels and pores. However, biological membranes and their associated proteins are challenging to experimentally characterize. These challenges have motivated recent advances in nanotechnology towards building and manipulating synthetic lipid systems. Liposomes—aqueous droplets enclosed by a bilayer membrane—can be synthesised in vitro and used as a synthetic model for the cell membrane. In DNA nanotechnology, DNA is used as programmable building material for self-assembling biocompatible nanostructures. DNA nanostructures can be functionalised with hydrophobic chemical modifications, which bind to or bridge lipid membranes. Here, we review approaches that combine techniques from lipid and DNA nanotechnology to engineer the topography, permeability, and surface interactions of membranes, and to direct the fusion and formation of liposomes. These approaches have been used to study the properties of membrane proteins, to build biosensors, and as a pathway towards assembling synthetic multicellular systems.

## 1. Introduction

Lipid and protein-bound compartments are found across all domains of life, from bacteria and archaea to eukaryotes [1]. They play a fundamental role in the evolution of biological complexity, keeping related molecules together so that beneficial mutations lead to preferential selection [2], enabling differentiation and thus specialisation [3]. However, they require methods to transmit information and material into and out of the compartment. For lipid-bound cells, membrane proteins are responsible for signal transduction, cell-to-cell communication, and the transport of ions and molecules into and out of the cell [4]. 

Membrane properties and interactions are of key significance in understanding a large range of biological processes and pathologies. Membrane-bound proteins (MPs) drive diverse biological processes and make up 23% of the human proteome [5]. They hold particular importance as drug targets in modern medicine, with 50% of drugs targeting only four key MP gene families [6]. Defects in MPs are implicated in many diseases, including Alzheimer’s [7] and cystic fibrosis [8]. More recently, it has become clear that membrane domains called lipid rafts form regions that are ordered and tightly packed, playing crucial physiological roles across cell types that range from immune cells to cancer cells [9]. Enzymatic regulation of the lipid composition itself has also been found to be involved in signal transduction and other vital cell processes [10]. 

The primary application of liposome technology is in drug delivery. By strategic design and synthesis of liposomes, it is possible to influence the timing and location of the release of a compound inside a patient, both to reduce the side effects and to increase the efficacy of a drug [11]. This encapsulation has proven therapeutic benefits, with 15 FDA-approved products on the market and 6 in phase 3 clinical trials in 2017 [12]. Technological developments in controlling lipid membranes have also driven significant advances in our understanding of biological membranes and membrane proteins, and synthetic lipid membranes act as models for cell membranes in which protein function can be tested. These methods have provided the ability to maintain MPs in their native state while solubilising them in aqueous environments for experimental characterization [13]. In parallel, in the field of DNA nanotechnology, DNA has been developed as a programmable building material for self-assembling biocompatible nanostructures for use as tools for biophysics [14,15], templates for nanofabrication [16], and platforms for diagnostics and therapeutics [17].

Recently, the fusion of DNA nanotechnology and synthetic lipid techniques has driven the development of new methods for manipulating lipid membranes and membrane proteins. Lipid-DNA nanostructures can act as biosensors [18], synthetic nanopores [19], scaffolds to produce non-spherical lipid tubes and spirals [20], and scaffolding for membrane proteins in biophysical experiments [21,22]. Lipid bilayers have also been used as a substrate for nanostructure assembly [23] and to encapsulate therapeutic DNA nanostructures, protecting them from enzymatic degradation [24]. 

Here, we give an overview of techniques from synthetic lipid technology (Section 2) and DNA nanotechnology (Section 3), and the limitations of each, which have motivated the fusion of the two. Hydrophobically modified DNA oligonucleotides, which can act to bridge these two areas, are then introduced (Section 4). Following this, we discuss in detail the recent work on two specific approaches: membrane-spanning (Section 5) and membrane-shaping (Section 6) DNA nanostructures. We end by summarizing some applications of this technology in controlling transmembrane signaling and controlling liposome fusion, and speculating on future directions.

## 2. Liposomes

### 2.1. Liposome Structure

Membrane bilayers are ubiquitous in nature and form the basis of biological compartmentalisation. Liposomes are synthetic vesicles with an aqueous core surrounded by one or more bilayers of amphipathic molecules, such as phospholipids. The similarity of liposomes to cell membranes makes them a powerful tool for modelling cellular membranes in simplified synthetic systems. First described in 1965 [25], liposomes can be easily and reproducibly produced in a variety of sizes, morphologies, and compositions. The biocompatibility and encapsulation ability of liposomes have led to their use in a range of applications as therapeutics and in industrial bioprocesses [26].

Liposomes can be classified on the basis of structural properties, such as size and lamellarity (Figure 1). Unilamellar liposomes consist of a single bilayer and are classified into three sub-categories: small unilamellar vesicles (SUVs; up to approximately 100 nm), large unilamellar vesicles (LUVs; between approximately 100 nm and 1 μm), and giant unilamellar vesicles (GUVs; larger than 1 μm) [27]. Multilamellar liposomes (MLVs) consist of many layers of concentric bilayers and are usually larger than 500 nm. Liposomes consisting of only a small number of concentric bilayers are termed oligolamellar vesicles (OLVs) and can be as small as 100 nm [28]. Liposomes enclosing either multiple non-concentric bilayers or multiple non-concentric interconnected monolayers are described as multivesicular liposomes (MVLs) [29,30]. Alternative liposome classification systems are based on the charge of lipids or functional chemical modifications [26]. 

Bilayer model systems can also be synthesized by pairwise fusion of droplet water-in-oil monolayers [13]. These systems allow for different orientiations of bilayers that enable mounting on microscopy for simultaneous electrophysiology recording and fluorescence measurements [31,32,33].

### 2.2. Lipid Chemistry

Lipids, the primary component of liposomes, are amphipathic or hydrophobic small molecules. Structural information is available for over 40,000 unique lipids [34]. Lipids are typically formed by condensations of thioesters or isoprene units and are divided into eight categories on the basis of their constituent chemical subunits: glycerophospholipids, glycerolipids, fatty acyls, polyketides, sphingolipids, sterol lipids, prenol lipids, and saccharolipids [35,36].

The primary function of a membrane is as a barrier. This barrier is in fact itself a highly non-homogeneous fluid that rests on a deformable manifold [37]. Lipid structure, specifically the saturation of the fatty acid tails, strongly influences the membrane ‘fluidity’ [38]. Fluidity describes the amount of disorder and the rate of lateral diffusion of molecules within a membrane [39]. Van der Waals interactions between lipid tails are the main force responsible for mediating lipid organisation. Double bonds contained within unsaturated fatty acids induce a kink in the structure of the tail, disrupting these interactions and increasing disorder. Hydrogen bonding interactions between lipid headgroups and tail length also affect fluidity [40].

Systems of lipids transition between a gel phase at cool temperatures and a liquid phase at warm temperatures. In the gel phase, lipid tails are normal to the membrane surface, are tightly packed, and show minimal diffusion. In the liquid phase, lipids vary in orientation, disperse, and more readily diffuse [41]. Changes in both lipid–lipid and lipid–water interactions at a given transition temperature affect tail orientation relative to the bilayer membrane surface and can induce a change in lipid fluidity. Lipid chemistry also influences membrane characteristics via interactions between headgroups and dissolved salts at the water–membrane interface, which affects the surface charge of the bilayer membrane [42].

### 2.3. Preparation of Synthetic Liposomes

A number of methods exist for producing and processing liposomes with control over size, lamellarity, and solution composition [43]. Common methods for producing multilamellar liposomes include hydrating a film of several layers of dried lipids [44]; evaporating lipid-containing solvent emulsion droplets from an agitated aqueous mix [45]; and adding proliposomes, lipids encapsulated in a salt or sugar granule, to an aqueous solution [46]. Multilamellar vesicles can be converted into unilamellar vesicles by extrusion and sonication [47].

Alternatively, unilamellar liposomes can be directly generated in several ways. These include detergent dialysis, where detergent-stabilised lipid micelles spontaneously form liposomes as the detergent is removed [48]; solvent injection, when lipids suspended in a solvent are injected a warm aqueous solution [49]; and reverse phase evaporation, where an agitated solvent containing dissolved lipids and water-in-oil emulsion droplets is gradually removed by evaporation [50]. 

Micron-scale unilamellar liposomes (GUVs) can be produced by methods such as natural swelling, where dried lipid films are slowly hydrated [51], and electroformation, where lipids on an electrode surface swell due to an electro-osmotic effect caused by an AC current [52,53]. Developments in microfluidics systems have led to liposome preparation methods such as hydrodynamic flow focusing, where a thin stream of lipids in alcohol diffuses into an aqueous medium [54], and pulsed jet flow, where droplets of aqueous solution are shot through a stabilised planar membrane [30,55].

### 2.4. Limitations of Liposome Technology

Improvements in methods to produce synthetic liposomses of controlled composition, position, and lamellarity have enabled the detailed mechanistic study of of purified membrane proteins. For example, the real-time simultaneous single-channel current recording and fluorescence microscopy observation of the mechanosensitive ion channel MscL, while undergoing force-activation, was made possible by recent advances in droplet hydrogel bilayers [32]. However, several limitations still exist. Current challenges include the specific functionalization, precise control over size, the ability to make non-spherical shapes, dynamic reconfiguration of liposome shape and size, and precise control over fusion. As introduced in the following section, DNA nanotechnology provides the ability to precisely control nanoscale structures and dynamics in biomolecular systems, and has the potential to address these limitations. 

## 3. DNA Nanotechnology

### 3.1. DNA Nanostructures

The molecule DNA has a number of properties that make it suitable for use as a building material for self-assembling nanostructures. It is a programmable polymer with strong, specific interactions based on Watson–Crick base pairing [56]. It is suitable for nanoscale construction with a diameter of ≈2 nm; a helical pitch of ≈3.4 to 3.6 nm; and has a combination of flexibility, with a persistence length of ≈4 nm for single-standed DNA (ssDNA) [57]; and rigidity, with a persistence length of ≈50 nm for double-stranded DNA (dsDNA) [58]. These factors, coupled with the ease of nucleic acid synthesis [59], automated design software [60], and the ability to conjugate both the DNA backbone and nucleobases with covalent modifications [61], have driven the field of DNA self-assembly since its inception over 30 years ago by Ned Seeman [62].

There are currently two design paradigms within structural DNA nanotechnology: DNA “origami”, which uses a long “scaffold” strand to nucleate assembly, and DNA “bricks” or “tiles”, which do not use a scaffold. DNA origami is one of the most versatile methods for assembling complex DNA nanostructures in two and three dimensions with high yield in a single step [63,64]. In this technique, a long piece of single-stranded M13 bacteriophage DNA (≈7000 nt) is folded into a compact shape, and cross-linked with shorter synthetic ‘staple’ strands. 2D origami were originally published in 2006 [63], and this technology was extended to folding shapes in three dimensions in 2009 [64]. DNA origami nanostructures typically have fast and robust assembly, high yield, and modular design, making them accessible to non-specialists. 

In comparison, in the DNA brick or tile method, many short synthetic DNA strands are tiled together into larger objects, without the nucleating viral scaffold [65,66]. Although more expensive and generally lower in yield, DNA brick structures have a greater diversity of possible sizes and shapes, and can also be used to build up crystalline lattices of DNA up to millimeters in size [67,68].

Using either DNA origami or DNA tile methods, it is possible to arrange DNA helices in a number of geometries. Some commonly used helix motifs include 2D planar “rafts” or “rectangles” of parallel helices [63], longitudinal “bundles” of helices [69], 3D blocks of DNA helices packed on either honeycomb or square lattice [64], 2 or 3D wireframe structures [70], and 2 or 3D structures built from stacked concentric rings of DNA [71]. However, the designs most widely used in many applications remain the 2D rectangular tile [63] and the six-helix bundle [69]. 

DNA nanostructures can also be decorated with other functional elements such as proteins [72], metallic nanoparticles [73], and therapeutic molecules [17], as well as by covalent modifications of component DNA strands. DNA origami have been used as a scaffold to organise individual carbon nanotubes into a transistor junction [74], to route single polymer chains that act as conducting nanowires into complex shapes in 2 and 3 dimensions [16], and to arrange gold nanoparticles for plasmonic structures [75]. DNA nanostructures have also been developed as in vivo drug delivery vehicles [17,24,72]. In the following section, incorporation of hydrophobically modified DNA strands into DNA nanostructures will be discussed in more detail. 

### 3.2. Switchable DNA Nanomachines

Alongside structural elements, a range of dynamic and environmentally responsive elements have been built from DNA [76]. One example is an autonomous DNA motor that chooses among four routes to navigate a maze of tracks to deliver a molecular cargo [15]. DNA molecules can also be designed to act as logic gates and process parallel molecular inputs in a programmable way, for example, to compute the square root of a number [77]. DNA strand displacement reactions underpin the majority of these functions. In DNA strand displacement, one DNA strand displaces another by bonding to an exposed single-stranded sequence known as a toehold [78]. Although the many diverse applications of strand displacement have been reviewed comprehensively elsewhere [78,79], an example relevant to lipid-interacting DNA nanostructures is the use of strand displacement to trigger conformational changes in DNA nanostructures [80]. 

Switching mechanisms that respond to environmental or external signals have been developed, for example, by incorporating pH sensitive i-motif structures [81] or salt-sensitive G-quadruplex structures [82]. Other environmental triggers such as photosensitive oligonucleotide modifications [79,83], and structures responsive to external electric [84] and magnetic [85] fields have also been demonstrated.

### 3.3. Limitations of DNA Nanotechnology

Similarly to lipid nanotechnology, DNA nanotechnology has allowed for the study of protein systems with unprecedented precision. For example, in recent work, a DNA nanospring was used as a force sensor to make direct single-step observations of the cellular protein motors Myosin V and Myosin VI under load, confirming for the first time that the stepping behaviour of these motors depends on the force applied to them [86]. However, this technology also has limitations. Current challenges include the stabilisation to prevent denaturation and degradation in serum, the ability to make fast and efficienct molecular machinery approaching the sophistication of protein motors, and the porosity of DNA nanostructure capsules to small molecules. As discussed in the following sections, the ability to chemically modify DNA strands with hydrophobic domains allows for the fusion of DNA and lipid nanotechnology, and provides the pathway to addressing some of these challenges. 

## 4. Hydrophobic DNA Modification for Membrane Attachment

### 4.1. Chemistry of Hydrophobic DNA Modifications

In order to fuse DNA and lipid nanostructures, we require a bridge. The chemical modification of DNA oligonucleotides with hydrophobic domains provides that bridge. A range of hydrophobic molecules have been conjugated to oligonucleotides to facilitate their attachment to membrane bilayers (Figure 2). These include cholesterol, tocopherol, porphyrin, polypropylene oxide, single chain fatty acids, steroids, and peptides [87,88,89]. The particular attachment most suitable for membrane insertion depends on the lipid composition of the membrane and the affinity of a conjugated molecule for a particular lipid domain [90]. For example, hydrophobic groups such as porphyrin can be used to optimise insertion efficiency and also allow fluorescent imaging [91], whereas modifications such as tocopherol can be used to target insertion into specific membrane phases (liquid disordered) [92]. 

### 4.2. Assembly Methods for Lipid-DNA Nanostructures

Hydrophobic modifications are typically incorporated into DNA nanostructures using one of two approaches. In the first, DNA nanostructures are folded with many hydrophobically modified staples to assemble structures decorated with many hydrophobic groups [95]. In the second method, hydrophobically modified ssDNA strands are first incorporated into a lipid bilayer to act as a “handle”, which then promotes docking of a DNA nanostructure decorated with a complementary “anti-handle” strand [96].

Alternatively, lipid molecules can be covalently linked directly to DNA origami. Maleimide-modified lipid molecules (PE, DOPE) have been conjugated to thiol-modified DNA staples [97,98]. This conjugation can be performed either on ssDNA staples before origami folding [98], or on folded DNA origami nanostructures [97]. Similarly, amphipathic peptides, such as the N-terminal helix of the ESCRT protein Snf7, have been directly conjugated to DNA staples for use as membrane anchors [99].

### 4.3. Properties of Different Lipid Modifiations 

The quantity, type, and position of hydrophobic molecules that decorate a DNA nanostructure will all influence lipid binding efficiency. When cholesterol anchors were moved from the centre to the periphery of a rectangular DNA origami structure, a 10-fold increase in membrane co-localisation was observed [95]. Spacers between hydrophobic groups and DNA nanostructures, such as tetraethylene glycol (TEG) or dsDNA, also increase binding efficiency by overcoming steric hindrance [100]. The chain-length of the membrane anchor is also relevant to binding strength and stability in the membrane. In DNA-mediated liposome fusion (Figure 3) the chain length was increased from decyl to solanesol, approximately doubling the length, such that the tail could span the full bilayer. This resulted in stronger binding and ≈3-fold increase in liposome fusion as measured by lipid transfer and content mixing [94]. Similarly, for construction of DNA-circled lipid nanodiscs, an increase in efficiency was seen with increasing chain length from ethyl to decyl [101].

### 4.4. Bridging DNA and Lipid Nanotechnology 

Functionalising DNA with groups that allow for incorporation into lipid bilayers allows the combination of lipid and DNA nanotechnology. This combined apparoch can achieve far more than either technology alone. For example, the precise structural scaffolding that is available via DNA nanotechnology can be used to precisely arrange lipid structures. Conversely, liposomes and membrane barriers allow the integration of membrane proteins with DNA nanotechnology, which allows for more complex and more specific chemical signaling, as well as coupling DNA computing to the encapsulation and delivery of a compound at a specific time and place. In the following two sections, we discuss in detail two specific areas of high interest in this field, membrane-spanning and membrane-shaping DNA nanostructures. 

## 5. Membrane-Spanning DNA Nanostructures

### 5.1. Towards DNA Nanostructure Transmembrane Signaling

Transmembrane proteins are the gateway to the cell, controlling influx and efflux of biochemical signals. Ion channels can selectively control flow of specific ions across a membrane and can switch ion transport on or off (i.e., be “gated”) in response to chemical, light, or mechanical stimuli. Sensory ion channels are the basis of touch, sight, and hearing. In tissues, multiple ion channels can interact to achieve sophisticated functions, such as the voltage and mechanosensitive ion-channels that maintain cardiac rhythm [106]. Such diverse and complex behaviours make transmembrane proteins attractive for adapation for other purposes. For example, protein nanopores have been adapted for use in single-molecule DNA sequencing [107]. These proteins also motivate the development of entirely synthetic DNA nanopores, which have the potential to combine the functional diversity of protein nanopores with the precise external control of DNA nanotechnology. 

A number of bilayer-spanning DNA nanostructures have been developed, which can be categorized into two broad types: (1) unscaffolded tile structures assembled out of short (<60 nt) oligonucleotides and (2) scaffolded DNA origami structures folded from M13-based scaffold and short oligonucleotide staples (Figure 3). A general feature of all designs is the presence of transmembrane DNA helices oriented perpendicularly to the membrane surface. The nanostructures interact with the membrane via hydrophobic modifications either along the length of the transmembrane region (Figure 3A–E) [103], or on raft-like regions that dock onto the surface of the membrane (Figure 3F–H) [19]. These features mimic the role of hydrophobic domains in transmembrane proteins [108]. 

### 5.2. Unscaffolded Membrane-Spanning DNA Nanostructures

Unscaffolded DNA nanopores include a single DNA duplex (Figure 3A), four-helix bundles (4 hb) (Figure 3B) and six-helix bundles (6 hb) (Figure 3C). In the simplest approach, a single DNA duplex decorated with six porphyrins was inserted into a lipid bilayer and induced ion conductance across the membrane of up to 0.1 nS [109] (Figure 3A). It was proposed that the ions did not cross the membrane through the duplex, but rather through a toroidal pore around the duplex that was generated as the lipid molecules rearranged around the duplex.

Larger transmembrane pores are expected to have larger conductance. An 11 nm tall 4 hb DNA nanopore, following the single-stranded tile (SST) motif [65], has an estimated outer diameter of 5 nm and inner diameter of 0.8 nm. The 4 hb was observed to to generate a higher ionic current across the bilayer than the single duplex, exceeding 0.3 nS (Figure 3B) [102]. In comparison, a 6 hb (Figure 3C) has an estimated outer diameter of 7.5 nm, and inner diameter of ≈2 nm [103]. Molecular simulation of a 6 hb inserted in a lipid membrane predicted conductance of 4 to 20 nS, depending on the salt concentration [110]. Experimentally, 6 hb conductance was generally lower than predicted, but higher than for 4 hb or a single duplex. For 15 nm and 9 nm tall 6 hb structures decorated with ethyl groups, average ionic conductance values of 0.395 ± 0.097 nS and 1.62 ± 0.09 nS were measured, respectively, whereas for a 14 nm 6 hb decorated with porphyrin, conductance of the high conductance state was measured as 1.62 ± 0.07 nS [111].

The internal diameter of a transmembrane 6 hb has also been experimentally estimated by the translocation of polyethylene-glycol (PEG) molecules of different sizes. PEG molecules have a roughly spherical shape, low reactivity, and predictable effects on the conductivity of membrane pores, and thus provide an effective method for estimating the interior diameter of lipid pores [112]. For 6 hb, PEG molecules above a cutoff of 1000 Daltons (hydrodynamic diameter of ≈1.9 nm) were not observed to translocate, which is consistent with the predicted internal diameter of 2.0 nm [111]. Molecules of this size are expected to be too large to “leak” through the 1 nm aqueous gap, which is proposed to form between the DNA channel structure and the surrounding lipid membrane [109,111].

### 5.3. Scaffolded Membrane-Spanning DNA Nanostructures

Scaffolded DNA origami allows for the assembly of DNA nanopores as part of larger DNA nanostructures. This allows a broader variety of shapes and a higher number of hydrophobic moieties. A barrel-shaped DNA origami has been used to dock a 42 nm tall 6 hb transmembrane DNA channel into a membrane via 26 cholesterol groups facing the membrane surface (Figure 3F) [19]. More stable membrane insertion was observed when cholesterols were placed closer to the central channel. An average conductance of 0.87 ± 0.15 nS was measured via single-channel electrophysiological measurements, which is of similar magnitude to 6 hb conductances for unscaffolded structures.

Scaffolded DNA origami designs allow for pores with larger internal cross sections, resulting in higher conductivity than the 6 hb and translocation of larger molecules across the membrane (Figure 3G,H) [92,105]. The DNA origami “T-pore” consists of a flat plate and a 27 nm tall 4 × 4 helix square channel with an estimated external width of ≈8 nm and internal width of 4.2 nm (Figure 3F) [92]. The plate is attached to the membrane either via 57 tocopherol groups, which were inserted into the membrane, or via 57 biotin-streptavidin groups, which bound to biotin-lipids present in the membrane. For the T-pore, translocation of ssDNA and dsDNA was observed, and an average conductance of ≈3.1 ± 0.3 nS was measured. 

Larger conductance (a mean of 30 nS) was observed for a funnel-shaped DNA origami, consisting of an 11 nm tall 5 × 5 helix square channel with an estimated external width of 10 nm and an internal width of 6 nm [105]. The second layer of the funnel was docked onto the surface of the membrane via 19 cholesterol groups. Very recently, a funnel-shaped scaffolded nanopore with a duplexed square lattice was assembled that could accommodate folded proteins at high flux and be driven beyond equilibrium [113]. 

Although increased numbers of hydrophobic modifications on a single DNA origami nanostructure result in better membrane insertion, they also promote nanostructure aggregation in solution. In all three examples above, the DNA origami structures were first folded with only plain staples, followed by incubation with hydrophobically modified ssDNA oligonucleotides. Generally, heating of cholesterol and tocopherol modified strands is required prior to addition of these strands to the DNA origami structures, and this incubation step is often done in the prescence of detergents. These additional steps add procedural complexity to the synthesis of the nanostructure.

### 5.4. Switchable DNA Lipid Channels

Switchable, or gated, ion transport in response to chemical, light, or mechanical stimuli is a functional feature of many membrane proteins. The diversity of switchable DNA nanostructures, which undergo a conformational change in response to local or external signals, provide a good basis for the development of DNA transmembrane nanostructures with similar functionality. 

Synthetic DNA channels also display gating proposed to be caused by thermal fluctuations of the structure [19]. Amplitude and timing of gating events was able to be influenced by the presence of 7 nt ssDNA strand protruding from the central channel, thus demonstrating the effect of structural details on electrical activity [19]. Molecular dynamic simulations have predicted fluctuations in the size of the opening at both ends of a 6 hb channel on the nanosecond time scale [114]. These simulations indicated that a DNA channel with an initial diameter of 2 nm could transiently constrict to 1.1 nm diameter below a simple threshold of 1.2 nm for an hydrated ion, and that the kinetics of these closing yielded signal-like traces for nanopore transitions [114].

The 6 hb channel has also been observed to switch between two distinct states of high and low conductance, dependent on voltage and topography. It adopted a low-conductance state more frequently at high voltages and when inserted into a nanopipette-mounted bilayer under small negative pressure [19]. Discrepancies between measurements of conductance across various publications have been proposed to be due to the measurement method, the applied voltage, and the chemical modification used for lipid anchoring [111]. Rational design of controlled gating has been demonstrated for 6 hb DNA-lipid channels. A switchable gating mechanism was demonstrated for the 9 nm tall 6 hb by incorporation of a blocking strand at one end of the channel (Figure 3D). This blocking strand could then be removed by displacement strand to open the channel and increase conductance from 0.66 ± 0.06 nS to 1.34 ± 0.08 nS [91]. Recently, this blocking strand technique was adapted to form a temperature-sensitive valve which was closed at ambient temperatures and open above 40 °C [115]. In another approach, a bridge of flexible ssDNA was placed across the end of the 6 hb channel (Figure 3E) [104]. Addition of a sequence complementary to the bridge ssDNA hybridized the bridge into more rigid dsDNA, putatively widening the end of the channel. This increased conductance from 0.3 ± 0.7 nS to 0.44 ± 0.14 nS and caused an increase in the rate of dye leakage [104]. The change in shape of the channel was verified by Förster resonance energy transfer (FRET). 

### 5.5. Challenges and Future Directions of Membrane-Spanning DNA Nanostructures

Despite much progress in the field of membrane-spanning DNA nanostructures, there remain challenges. There is currently a large degree of uncertainty regarding incorporation efficiency, permeability, and orientation within the membrane [111]. Variability in electrical conductance has been observed in a number of studies of DNA nanopores [87]. Molecular dynamics studies suggest that the walls of DNA channels are porous to ions and water, unlike those of protein nanopores [114]. Variability has also been observed between different DNA nanopore designs [87], even in the same channel interrogated using different electrophysiology techniques [111]. Thus far, the orientation of a DNA nanopore within a membrane has not been directly measured, but a more recent design achieved reliable and predictable conductance by stabilising the pore’s insertion with a large membrane-bound tile [92].

The energy penalty from inserting negatively charged DNA nanostructures through the hydrophobic core of lipid bilayers limits efficient and stable insertion. This has often required high concentration of nanostructure and the application of external voltage [19,116]. The required energy for insertion can be estimated using coarse-grained models such as MARTINI force field, developed to simulate lipid bilayers and their interactions with a range of biomolecular structures [117,118]. Using modelling and all atom simulations, the free energy gain from the insertion of a cholesterol can be predicted [119]. However, unlike coarse-grained models, all-atom simulations are limited in their capacity to model long time scales, such as the insertion of a DNA nanopore into a lipid bilayer. They have proven useful in simulating DNA nanopore behavior, once inserted, in assessing stability and in modeling the movement of ions through and around the pore to predict its electrical properties [105]. Likewise, molecular dynamics simulations have provided critical insight into the organizational principles of cell membranes [120]. 

## 6. Membrane-Shaping DNA Nanostructures

### 6.1. Bio-Inspired Membrane Shaping 

Control of membrane curvature and fusion plays an essential role in cellular trafficking, signaling, and function [121]. Many proteins sense, respond to, or direct changes in curvature [122]. However, there is an energetic cost to forming non-spherical lipid bilayers, and an array of lipid, protein, and cytoskeletal components are required [121]. For example, BAR (Bin/amphiphysin/Rvs) proteins electrostatically bind to membranes and oligomerise to induce concave or convex curvature [123,124]. Similarly, the dynamin, ESCRT (endosomal sorting complexes required for transport) and clathrin proteins are involved in directing membrane curvature in vesicle formation and endocyotosis [10,89,125]. Synaptic vesicle fusion is mediated by SNARE (soluble *N*-ethylmaleimide sensitive factor attachment protein receptor) proteins on the two membranes, which dimerise and zip up to promote membrane fusion [126].

Membrane-binding DNA nanostructures have been developed to mimic the roles of proteins in controlling membrane curvature [99,127] and fusion [128]. These structures have been used for a range of applications. Control of bilayer shape has been used to wrap DNA nanostructures in a protective bilayer to prevent nuclease digestion and increase in vivo circulation time [24] and to assemble 2D DNA lipid ‘nanodiscs’ for the study of membrane protein interactions [129]. DNA-guided fusion has been used to bypass the endosomal pathway for delivery of proteins into live cells (Figure 4A) [130] and to assemble biosensors for microRNA detection (Figure 4B) [131]. Lipid-docking DNA nanostructures have been used to scaffold defined numbers of SNARE proteins to test the effect of cooperativity when producing liposomes of well-defined size [21] and to characterise cell membrane properties such as diffusion and disorder [132]. 

### 6.2. Deforming Liposomes

3D DNA origami curved helix-bundles, designed to mimic BAR proteins, were decorated with cholesterol on either concave or convex face and observed to deform membrane topography and vesicle shape [97]. This method was able to produce lipid nanotubes decorated with DNA origami. However, semi-circular DNA nanostructures of tight curvature were not able to bind, suggesting that the high energy cost of creating tight bends in the membrane puts a limit on the degree of curvature that can be created with this technique. Short, curved, 24-helix DNA origami rods, modified with amphipathic peptides on the concave face, were oligomerised into a left-handed helix to mimic ESCRT proteins. These nanostructrues deformed large unilamellar vesicles into tubular shapes and created long tubular membrane protrusions on giant unilamellar vesicles (>250 μm) [99]. Recently, curved spring-shaped DNA origami structures were used to promote membrane binding and remodeling upon triggered polymerization [99]. DNA origami curls of varying thickness and curvature that were decorated with varying densities of amphipathic peptides were assembled and, by adjusting the number of membrane anchors, the diameters of lipid tubules could be controlled [99].

DNA origami monomers can also be assembled into larger lattices on membrane surfaces. 3D DNA origami block-shaped tiles decorated with cholesterol groups were designed to self-assemble into larger arrays on the surface of GUVs, distorting their shape [132]. Three-armed 3D DNA orgami were designed to mimic the triskelion-shaped clathrin, a key endocytic protein, and were shown to polymerize into a lattice on the membrane surface in a process similar to the formation of clathrin-coated pits [134]. The formation of this lattice was found to inhibit fusion of the DNA-coated GUVs with a supported lipid bilayer. 

### 6.3. Directing Liposome Formation

DNA nanotechnology can also be used to direct the formation of vesicles. Detergent-solubilised lipids spontaneously assemble into bilayers on removal of the detergent by dialysis. Hypdrophobically-modified DNA rings or cages can provide a template for bilayer formation during this process. When cholesterol modifications were placed on the outside of a spherical DNA cage, the lipid was observed to wrap around and protect the nanostructure [24]. However, this method requires PEG-modified lipids to prevent aggregation of lipid-wrapped nanostructures, and has not currently been shown to work for non-spherical shapes. In contrast, when the modifications are placed on the inside of the DNA cage, a range of shapes and size liposomes can be made. For example, a DNA origami ring was used to direct formation of vesicles of precisely controlled size, and the DNA ring was then removed by DNAse digestion [98]. 3D DNA origami cage structures of closely stacked rings were used to form tubular, toroidal, and helical liposomes [20]. For these structures, DNA handles or functional modifications can be added to the exterior of the DNA nanostructure template to purify or add function to the liposome.

Following a similar process, DNA nanostructure templates can be to direct the formation of two-dimensional bilayers. DNA circles of small diameters (approximately 15 nm) with an inward facing hydrophobic surface of alkylated nucleotides direct the assembly of very small lipid bilayer discs surrounded by a DNA frame [101]. Multiple copies of these DNA-encircled bilayers can then be scaffolded into a larger DNA ring and fused by modulating detergent concentration to make a large, continuous bilayer of up to 70 nm. Large, DNA-supported 2D bilayers can also be created using this technique by scaffolding and fusing multiple protein lipid nanodiscs inside a DNA origami barrel [129].

### 6.4. Controlling Fusion

To achieve fusion, spherical liposomes must be transiently deformed into non-energetically favoured states. In biological systems, proteins such as SNARE mediate liposome fusion [21,135]. It is possible to use DNA hybridization to promote fusion by incorporating cholesterol-conjugated DNA strands into liposome membranes [133]. When two liposomes decorated with complementary DNA strands are brought close together, the DNA strands hybridise promoting fusion (Figure 4A). This system can be modified to include a hairpin that opens only in the presence of a microRNA target, which acts as a trigger for DNA hybridization and vesicle fusion (Figure 4B) [131].

In synthetic systems, it is possible to incorporate both DNA lipid tethers to promote liposome docking, and SNARE proteins to facilitate fast membrane fusion, for example, by assembling two sets of liposomes, each with a t-SNARE or v-SNARE and a complementary DNA lipid tether (Figure 4C) [128]. The DNA-lipid tethers can be tuned to control the distance between the liposomal membranes to maximise the fusion rate, with a maximum rate for a 40 bp (≈13 nm) tether. DNA origami rings can also be used to study the co-operation between SNARE proteins during protein-directed membrane fusion (Figure 4D) [128]. By controlling the orientation and number of SNAREs per vesicle, it was found that 1–2 pairs of SNAREs were sufficient for fast membrane fusion.

### 6.5. Functionalising and Characterising Membrane Surfaces

Membrane-bound DNA nanostructures can be used as tools to functionalise and characterise cell membranes in biological systems. DNA origami tiles can be attached and detached from lipid membranes, as well as programmed to oligomerise into large islands on the membrane surface [136]. These membrane-bound DNA origami tiles can be used for the reversible tethering of living cells. A cell in solution, hydrophobically labeled with cholesterol ssDNA, can be attached to a stationary anchored cell containing a DNA tile with a complementary ssDNA. The rate of integration of cholesterol-conjugated oligos into a membrane varies between cell culture types, likely as a result of their differing lipid composition [96].

DNA nanostructures have been used as probes for characterising the biophysical characteristics of lipid membranes. A 2D lattice of membrane-bound DNA tiles was shown to have salt- and heat-dependent phase selectivity on the membrane surface and demonstate self-oligomerisation [137]. A rigid 422 nm tall six-helix DNA rod with TEG-cholesterol anchors was used to model diffusion behaviour in GUVs. Perferential binding of the DNA rods to liquid disordered phases was observed in low magnesium concentrations [138]. DNA rods can also be used to differentiate lateral and rotational particle diffusion in membranes by comparing the motion of fluorophores attached to the centres and ends of the rods [139].

DNA nanostructures can also be used to influence lipid reactions between liposomes. DNA origami nanostructures have been used to organise liposomes at precise distances, confirmed via FRET, and then facilitate lipid transfer between liposomes over large distances [140]. Lipid transfer using the synaptotagmin-like mitochondrial lipid-binding protein (SMP) domain of extended synaptotagmin 1 (E-Syt1) could occur in the presence of the DNA nanostructure over a distance greater than that of the SMP dimer on its own [140]. 

## 7. Applications and Future Directions

Membrane-spanning and surface-bound DNA nanostructures have potential applications for engineering liposomes for drug delivery. Liposomes have been used for encapsulating therapeutic payloads in order to increase a drug’s circulation time, decrease toxicity, and focus its distribution to a targeted site [12]. To achieve this goal, there has been considerable interest in engineering liposomes to release a payload in response to stimului such as heat [141], light [142], bond to a particular tissue type [143], or to deliver a payload to a cells cytoplasm [144]. The ability to integrate functional DNA nanostructures such as switchable nanopores or surface-bound tiles bearing chemical modifications presents a powerful and promising pathway to engineering liposomes for therapeutic applications.

As well as a tool for modifying liposomes for drug delivery, DNA nanopores themselves have potential cytotoxic applications. Membrane-spanning DNA nanostructures have demonstrated a cytotoxic effect when added to cancer cell culture [145], analogous to the function of pore-forming toxins causing cell death by disrupting membrane integrity [146]. DNA nanopores have also been modified with a tumour cell-recognising DNA aptamer to promote selective adhesion to tumor cells [147].

Lastly, the rational design of DNA nanopores is a promising avenue for development in resistive pulse sensing, where the passage of an analyte through a narrow aperture is detected by a transient drop in voltage [148]. Translocation of DNA [19], proteins [113], and various small molecules [92,111] has been achieved, demonstrating the potential versatility of the technology in the sensing of biological molecules.

## 8. Conclusions

The combination of DNA and lipid technology offers the potential to exert molecular-level precision control over the environment of membrane proteins, as well as to make unprecendented measurements of their biochemistry and electrophysiology. DNA can provide the required stoichiomteric, geometric, and temporal control, whereas liposomes situate this in a biologically compatible environment. Building on recent advances, these technologies will continue to integrate further. There is even potentially a scope to reverse their traditional roles, where DNA acts as a geometrical scaffold and lipids as a biological mediator, for example, with hydrophobically modified DNA nanostructures acting like lipids and assembling at phase interfaces [149], and with lipids that are able to act as scaffolds for the spatial arrangement of DNA logic circuits [150] and protect them from degradation [151]. By integrating the latest in protein nanotechnology, DNA scaffolds have also been used to synthesise protein pores that only assemble when the requisite DNA is present [152]. Future applications that combine lipid, protein, and DNA will increase functionality and complexity offered by in vitro systems. These systems may soon accurately mimic complex multicellular biological processes and enable new in vitro assays based on bottom-up construction of complex organelles using synthetic biology. 

## Figures and Tables

**Figure 1 genes-10-01001-f001:**
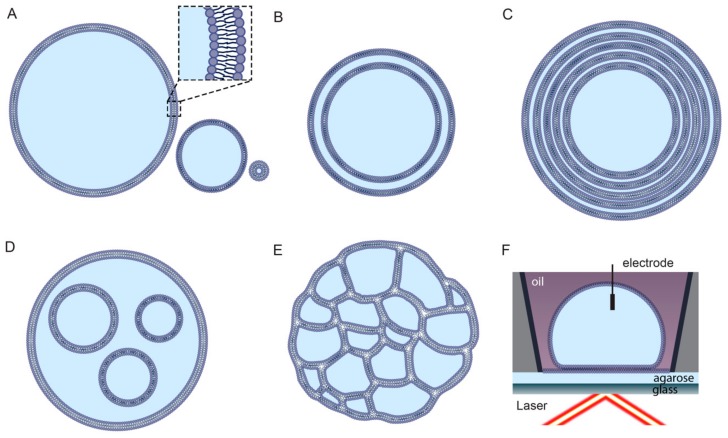
Schematics of droplet bilayers. (**A**) Unilamellar liposomes, or vesicles that are giant (GUVs), large (LUVs), or small (SUVs), respectively. Inset shows detail of two leaflets indicating lipid head and tail. Hydrophobic membrane interior is formed by fatty-chain lipid tails, with head groups in aqueous environment (blue). (**B**) Oligolamellar liposome (here a double bilayer). (**C**) Multilamellar concentric liposomes. (**D**) Multivesicular liposomes formed with bilayer-bound vesicles. (**E**) Multivesicular liposomes formed with monolayer-bound vesicles. (**F**) Droplet hydrogel bilayers. Controlled assembly of two lipid monolayers allows formation of bilayers of specific geometry and orientation for combined electrophysiology and fluorescence microscopy.

**Figure 2 genes-10-01001-f002:**
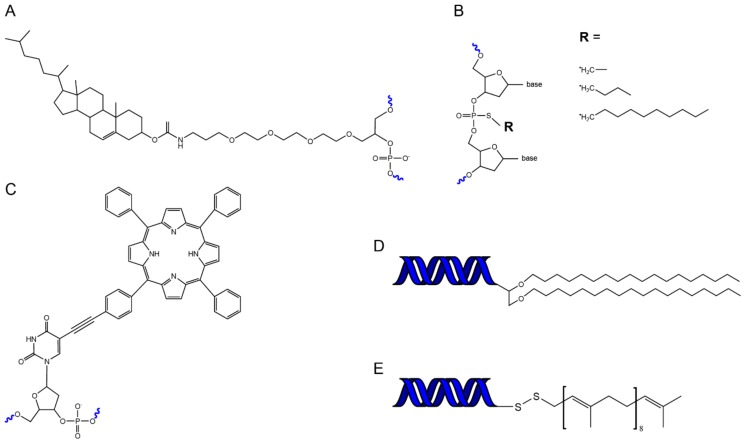
Chemical schematics for hydrophobic functionalized DNA oligomers. (**A**) Cholesterolated DNA. (**B**) Alkylated DNA. (**C**) Porphyrin-DNA [93]. (**D**) Diglycerol ether DNA [94]. (**E**) Solanesol DNA [94].

**Figure 3 genes-10-01001-f003:**
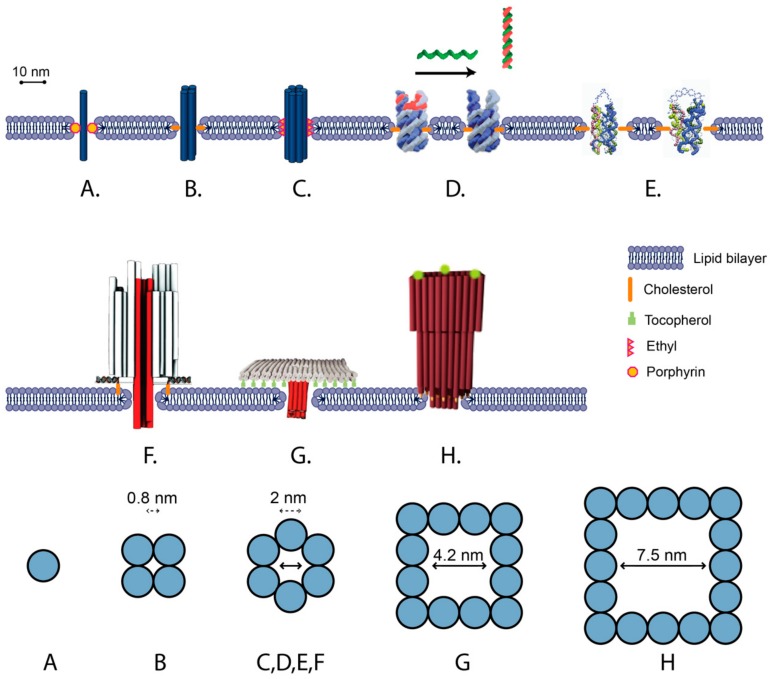
Summary of published membrane spanning DNA channel designs and attachment chemistries. The bottom row shows channel cross-section for each design A–H. (**A**) Channel made from single membrane-spanning DNA duplex decorated with six porphyrin tags. (**B**) Channel made from four-helix-bundle attached to membrane via cholesterol binding to the interior of the bilayer [102]. (**C**) Channel consisting of six-helix-bundle attached to membrane via ethyl groups binding to interior of the bilayer [103]. (**D**) Switchable six-helix nanopore that features a “lock” strand at upper opening that can be displaced on addition of complementary DNA oligonucleotide strand. Channel is attached to bilayer via cholesterol on the side of the six-helix bundle into the interior of the membrane bilayer [91]. (**E**). Switchable channel attached via cholesterol into the interior of the membrane. Upon addition of a single stranded DNA strand (ssDNA) that is complementary to the ssDNA strand at the opening, a rigid helix is formed which alters the conformation of the pore to a higher conductance state [104]. (**F**) Channel consisting of six-helix-bundle attached using cholesterol binding to the top side of the membrane. Origami model adapted from [19]. (**G**) Channel consisting of 12 helices attached via tocopherol from a planar DNA raft into the top side of the membrane [92]. (**H**) Funnel-shaped channel with large conductance with 16 helices at pore attached via 19 cholesterols into the top side of the membrane [105].

**Figure 4 genes-10-01001-f004:**
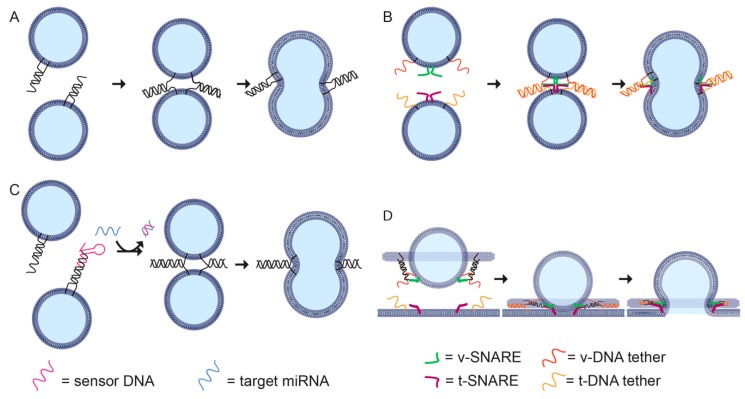
DNA nanotechnology-guided membrane fusion. (**A**) Membrane fusion by DNA tethers [133]. The DNA tethers on the liposomes contain complementary ssDNA that hybridize in a zipper-like manner, bringing the liposomes in contact and promoting fusion. (**B**) Acceleration of SNARE-mediated membrane fusion by DNA lipid tethers [131]. Two sets of liposomes containing v-SNARE (green) and t-SNARE (magenta) are brought together using complementary DNA tethers, v-tethers (red strand) and t-tethers (orange strand), respectively. (**C**) MicroRNA-specific membrane fusion [128]. Target miRNA (blue strand) hybridises to the hairpin (red strand) and displaces it, revealing a binding region for DNA tethers (black). DNA tethers on the liposomes can now hybridise in a zipper-like manner and induce membrane fusion, as in panel A. (**D**) Docking of a DNA origami-templated SUV containing v-SNAREs and v-tethers onto a supported lipid bilayer (SBL) containing t-SNAREs and t-tethers [128]. The hybridisation of the tethers results in the docking of the liposome and brings the SNARE proteins together, which in turn fuses the liposome with the SBL.

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
