# Peer review of "The Fusion of Lipid and DNA Nanotechnology"

_genes, 2019, doi:10.3390/genes10121001_

Round 1

Reviewer 1 Report

The manuscript entitled “The fusion of lipid and DNA Nanotechnology” discusses techniques from lipid and DNA Nanotechnology for engineering the topography, permeability and interactions of membranes. This review is a novel amalgamation of lipid and DNA Nanotechnology and presents the readers with an in-depth and elaborate knowledge of lipid chemistry and synthesis along with presenting to them an overview of DNA nanotechnology. Although the review covers some of the important topics for bringing forth the idea of how the merging of these two technologies is advantageous.

Nevertheless, in my opinion the presentation is not very concise and lacks to provide a clear and structured picture of the current state of art. Furthermore, the figures included are not very descriptive and well structured.

Comments:

In my opinion, a brief description of how these fields complement and advance each other should be included after the descriptions about both the technologies. It will help in understanding of the text better and provide the required structure which is missing in the review. Although, there is a brief mention about this in the introduction but it seems like a good idea to include more on this topic. In my opinion, the continuity among paragraphs is missing, there is an abrupt change of topic from one heading to the other, if the paragraphs can be presented in a more linked and organised fashion, the readers can better relate to the text and the review can become more comprehensive. Page 6 Line 202. 3.3 Hydrophobic DNA modifications for membrane attachment links both these technologies and should be more clearly explained with related examples from literature. Also, in my opinion it is better to have this as a separate subheading than to merge it under the heading of DNA nanotechnology, as the topic mentioned is the linking topic between both the technologies. The text under each subheading can be made bit more concise and structured, moreover addition of a separate topic for existing applications of both membrane shaping and membrane spanning with relevant examples from literature would make the review more interesting to read and comprehend. Although Page 10, line 375 -381 does mention some relevant applications, a separate and elaborate mention seems more suitable. Also, including existing challenges and complications with a brief idea of how to overcome them before the conclusion would be apt rather than moving on to conclusion hastily. In my opinion, the conclusion seems haphazardly structured, there should be a clear direction to it along with a brief description of future scope. More representative and self-descriptive figures should be provided to support the text. Page 2, Line 48 “the primary use of liposome technology is in medical technology especially for drug delivery “ Page 9, Line 321 Topic 4.4 Determining DNA lipid – channel dimensions and 4.5 Controlling the flow of ions through DNA- lipid channel should be placed under a different subheading as it is not another type of membrane spanning DNA nanostructure. In my opinion, mentioning the role of computational approaches in both the fields and their integration for future applications is essential.

Author Response

We thank the reviewers for their considered comments and time to improve our manuscript.

We respond to comments below.

Reviewer #1:

In my opinion, a brief description of how these fields complement and advance each other should be included after the descriptions about both the technologies.

We thank the reviewer for this suggestion and have added an additional paragraph at the end of Section 3 that highlights the complementarity of these technologies. Prior to this we have added a section on the limitations of each technology (Section 1.4 & 2.3) to better show how the combination of technologies can address some of these limitations.

In my opinion, the continuity among paragraphs is missing,

We have added connecting text to improve the flow of the article and improve links between each paragraph. In addition we have added and improved subheading labels to try and better maintain continuity.

Page 6 Line 202. 3.3 Hydrophobic DNA modifications for membrane attachment links both these technologies and should be more clearly explained with related examples from literature.

Also, in my opinion it is better to have this as a separate subheading than to merge it under the heading of DNA nanotechnology, as the topic mentioned is the linking topic between both the technologies.

We have restructured this section (now Section 3.3 Properties of different lipid modifications) and, as above, improved links between the technologies.

The text under each subheading can be made bit more concise and structured, moreover addition of a separate topic for existing applications of both membrane shaping and membrane spanning with relevant examples from literature would make the review more interesting to read and comprehend.

 We have altered subheading titles as suggested to improve comprehension.

Although Page 10, line 375 -381 does mention some relevant applications, a separate and elaborate mention seems more suitable.

We have now added a separate subheading for applications, Section 6 “Applications and future directions”, with additional examples of applications (L501-522)

In my opinion, mentioning the role of computational approaches in both the fields and their integration for future applications is essential.

We have added text specifically surrounding modelling and the integration of these methods with a view to future applications, modeling of DNA nanopores, and modeling of cell membrane organisation (L380-L391).

Also, including existing challenges and complications with a brief idea of how to overcome them before the conclusion would be apt rather than moving on to conclusion hastily.

We have created a new subheading for challenges and complications, Section 4.5 (Lines 368-378). There we discuss caveats and some of the existing challenges in more detail.

In my opinion, the conclusion seems haphazardly structured, there should be a clear direction to it along with a brief description of future scope.

We agree with the reviewer’s concern regarding the conclusion and thank the reviewer for raising this. We have reduced its length and revised the conclusion extensively. With the addition of an additional caveats section the conclusions have been made more concise and structured around the synergy between lipid and DNA nanotechnologies and future incorporation of other nanotechnology.

Reviewer 2 Report

The article “The fusion of lipid and DNA nanotechnology” by Es Darley et al. presents a comprehensive review on nanotechnology based on lipid structures, DNA structures and the combination of both. It is clearly segmented into four parts. The first one deals with lipid nanotechnology, the second with DNA superstructures such as DNA origami, the third with DNA structures spanning the cross-section of lipid membranes, the fourth with DNA interactions influencing the lipid membrane shape. Actually, only the last two sections keep the promise which is made in the title of the article (fusion of lipid and DNA approach).

Altogether, the article presents a valuable introduction into the complex field of lipid / DNA nanostructures. With this, the article deserves to be published in GENES. There are only a few minor points which should be addressed by the authors in order to improve the manuscript:

Line 13: “(193 words) should be deleted All figures lack some resolution, but maybe that is a problem of the draft version Lines 107 to 110: The authors mention eight categories, but list only seven. Line 255; better “ i.e.” or “ie” Line 274: “The design … is…” Line 423: “Hydrophobically…” Line 490: “… operate …”

Otherwise, I would support the acceptance of the article.

Author Response

We thank the reviewers for their considered comments and time to improve our manuscript.

Reviewer #2:

Line 13: “(193 words) should be deleted.

Done.

All figures lack some resolution, but maybe that is a problem of the draft version.

Figure resolution will be corrected with editors in final proofs. We have all figures in full resolution vector graphics for final publication.

Lines 107 to 110: The authors mention eight categories, but list only seven.

We have added the missing lipid category added to complete list

Line 255; better “ i.e.” or “ie”

Done.

Line 274: “The design … is…”

Fixed.

Line 423: “Hydrophobically…”

Added ‘hydrophobically-labelled’.

Line 490: “… operate …”

This line has now been removed in the edited conclusions.

Round 2

Reviewer 1 Report

I think the authors revised manuscript satisfactory.

Some improvement in spellings and English grammar would make this a good paper and good contribution. Some corrections are listed below:

Line 35 : After the abstract, the heading INTRODUCTION is missing

Line 213 : DNA motor that chooses among four routes to (navigate ) a maze of tracks to deliver molecular cargo

Line 226: Switchings mechanisms – Switching mechanisms

Line 283: approach

Line 644: stoichiometric